# The Glutaminase-1 Inhibitor [^11^C-carbony]BPTES: Synthesis and Positron Emission Tomography Study in Mice

**DOI:** 10.3390/ph16070963

**Published:** 2023-07-05

**Authors:** Yiding Zhang, Katsushi Kumata, Lin Xie, Yusuke Kurihara, Masanao Ogawa, Tomomi Kokufuta, Nobuki Nengaki, Ming-Rong Zhang

**Affiliations:** 1Department of Advanced Nuclear Medicine Sciences, National Institutes for Quantum Science and Technology, 4-9-1 Anagawa, Inage-ku, Chiba 263-8555, Japan; 2SHI Accelerator Service, Ltd., 7-1-1 Nishigotanda, Shinagawa-ku, Tokyo 141-0031, Japan

**Keywords:** [^11^C-carbonyl]BPTES, PET, glutaminase-1 (GLS1), glutaminase-1 inhibitor, senescence, [^11^C]acylation

## Abstract

Bis-2-(5-phenylacetamido-1,3,4-thiadiazol-2-yl)ethyl sulfide (BPTES) is a selective inhibitor of glutaminase-1 (GLS1), consequently inhibiting glutaminolysis. BPTES is known for its potent antitumor activity and plays a significant role in senescent cell removal. In this study, we synthesized [^11^C-carbonyl]BPTES ([^11^C]BPTES) as a positron emission tomography (PET) probe for the first time and assessed its biodistribution in mice using PET. [^11^C]BPTES was synthesized by the reaction of an amine precursor () with [^11^C-carbonyl]phenylacetyl acid anhydride ([^11^C]**2**), which was prepared from [^11^C]CO_2_ and benzyl magnesium chloride, followed by in situ treatment with isobutyl chloroformate. The decay-corrected isolated radiochemical yield of [^11^C]BPTES was 9.5% (based on [^11^C]CO_2_) during a synthesis time of 40 min. A PET study with [^11^C]BPTES showed high uptake levels of radioactivity in the liver, kidney, and small intestine of mice.

## 1. Introduction

Bis-2-(5-phenylacetamido-1,2,4-thiadiazol-2-yl)ethyl sulfide (BPTES, Figure 1) [1], which is a specific and selective inhibitor of glutaminase-1 (GLS1), has potent antitumor effects [2,3,4,5]. Previous studies reported that BPTES inhibited pancreatic invasive ductal adenocarcinoma cell proliferation, and preferentially slowed the growth of mutant isocitrate dehydrogenase 1 cells without inducing apoptosis [2]. BPTES showed clear synergistic antitumor effects with 5-fluorouracil in the human alveolar adenocarcinoma cell line A549 and the non-small-cell lung carcinoma cell line EKVX, resulting in significant inhibition on EKVX and A549 and on most of the NSCLC cell lines [3,4]. In addition, BPTES was shown to have wide pharmacological effects on diseases related to immunometabolism, inflammation, and fibrosis [6,7,8,9].

GLS1 is a necessary gene for the survival of human senescent cells [10]. Overexpression of GLS1 enhances glutaminolysis in the senescent cells and induces excess ammonia production, which neutralizes the lower pH and improves the survival of the senescent cells [10,11,12]. Thus, inhibition of GLS1 could offer a promising strategy for removing senescent cells (senolysis), which is beneficial for improving wide age-associated pathologies [10,11,12,13].

Recently, it has been reported that BPTES can remove senescent cells and ameliorate age-associated organ dysfunction, by inhibiting glutaminolysis [14]. BPTES has improved age-associated pathologies of geriatric diseases and lifestyle-related diseases by selectively removing senescent cells in aged mice [14,15,16]. In addition, it has been observed that BPTES rejuvenates human skin via the clearance of senescent cells [17].

Although BPTES showed significant inhibition of tumor growth and the removal of senescent cells, its poor solubility in water and organic solvents has decreased its bioavailability and limited its clinical development and animal application [18]. To improve the bioavailability of BPTES, many BPTES analogs have been synthesized [19,20,21,22]. Further, BPTES nanoparticles (BPTES-NPs) were developed to considerably attenuate tumor growth in certain models [18,23,24,25]. Therefore, accurately measuring the concentration of BPTES itself as a lead compound in targeted sites or organs and determining its pharmacokinetics in vivo helps evaluate the pharmacological effects of BPTES and its analogs and NPs. 

Positron emission tomography (PET) imaging is useful for monitoring molecules radiolabeled with short-lived positron-emitting nuclides (for example, ^11^C, half-life of 20.4 min; ^18^F, half-life of 109.8 min) in living animals and humans. PET is a diagnosis biomarker for various diseases [26,27], and an assessment tool for pharmacokinetics/pharmacodynamics in living animals and for drug development [28,29,30]. Notably, PET has a high sensitivity, and the injected radioactive probe in trace amount is sufficient even for high-quality images [31].

The objective of this study was to develop a route for the radiosynthesis of [^11^C-carbonyl]BPTES ([^11^C]BPTES, Figure 1) and to characterize its in vivo dynamic biodistribution using PET. Herein, we report the radiosynthesis and PET imaging of [^11^C]BPTES in mice to investigate the biodistribution of BPTES for the first time. As a proof-of-concept study, a PET imaging study using [^11^C]BPTES would provide valuable insights into the antitumor and senolytic effects of BPTES and the new BPTES analogs, or BPTES-NPs.

## 2. Results and Discussion

### 2.1. Synthetic Route

Considering its chemical structure, we labeled the carbonyl group in BPTES with ^11^C. (Figure 1). The labeling for this chemical functional group does not change the structure or pharmacological profiles of BPTES. The radiosynthesis of [^11^C]BPTES involves efficient construction of an [^11^C]amide moiety, which uses an amine precursor, N-(5-(2-((2-(5-amino-1,3,4-thiadiazol-2-yl)ethyl)thio)ethyl)-1,3,4-thiadiazol-2-yl)-2-phenylacetamide [1], and requires reliable preparation of [^11^C]phenylacetyl acid anhydride ([^11^C]**2**) (Figure 1).

### 2.2. Chemical Synthesis

Figure 2 shows the synthetic route of unlabeled BPTES and the precursor **1** for radiolabeling. The reaction of thiodipropionic acid **3** with thiosemicarbazide **4** in the presence of phosphorus oxychloride produced bis-aminothiadiazole **5** in a 14.2% yield [19]. The coupling of **5** with excess phenylacetyl chloride afforded a mixture of bi-(BPTES) and mono-(**1**) phenylacetylated products, which were separated by column chromatography to afford BPTES and precursor **1** for radiolabeling in 18.0% and 11.8% yields, respectively.

### 2.3. Radiosynthesis

The radiosynthesis of [^11^C]BPTES (Figure 3) was carried out using an automated synthesis system developed in-house [32]. Using this system, we have achieved a Grignard reaction with [^11^C]CO_2_ and the following [^11^C]acylation of amine or phenol precursors to produce PET probes with high molar activity (>37 GBq/μmol) and reproducible reaction efficiency and radiochemical yields [33,34,35,36,37]. The cyclotron-produced [^11^C]CO_2_ was firstly used to react with benzylmagnesium chloride, which was cooled in a polyethylene loop between −5 and 0 °C in advance, to produce [^11^C-carbony]phenylacetate. Treatment of the reaction mixture with isobutyl chloroformate (ClCOOiBu) produced [^11^C]phenylacetyl acid anhydride ([^11^C]**2**) in situ. Without separation of [^11^C]**2**, this radioactive reaction mixture was directly treated with precursor **1** to give a [^11^C]acylated mixture, which was purified by semi-separative HPLC to produce [^11^C]BPTES as an injectable solution for quality control and animal evaluation (Figure 4).

After the one-pot, three-step reactions, HPLC separation, and formulation, [^11^C]BPTES was obtained in 9.5 ± 2.3% radiochemical yield based on [^11^C]CO_2_ (*n* = 20), with sufficient radioactivity and reliable quality for animal evaluation. Starting from 25 ± 3.4 GBq of [^11^C]CO_2_, 370–550 MBq of [^11^C]BPTES was obtained at the end of synthesis. The total synthesis time was 40 min from the end of the bombardment. The radiochemical purity of [^11^C]BPTES was greater than 96% (Figure 4), and the molar activity was 50 ± 26 GBq/μmol, with a logD_7.4_ value of 3.59. The radiochemical purity of [^11^C]BPTES was > 95% after standing for 90 min at room temperature, indicating the radiochemical stability of this product for the period of one PET scan. The analytical results complied with the quality control/assurance specifications of radiopharmaceuticals produced in our facility.

### 2.4. Small-Animal PET Imaging

To evaluate the biodistribution of [^11^C]BPTES in vivo, mice were subjected to dynamic PET scans.

Figure 5 shows the typical static PET images of a mouse injected with [^11^C]BPTES. One minute after the injection, the injected radioactivity was carried through the vena cava to the heart. Radioactivity is immediately distributed throughout the body and appears in the lungs, heart, liver, and kidneys. The uptake in the lungs and heart peaked rapidly and was then cleared from them. Radioactivity accumulation in the urinary bladder indicates rapid and significant excretion in the urine. High radioactivity in the liver, gall bladder, and small intestine from the early period suggests biliary excretion of radioactivity. Radioactive concentrations in the liver and other tissues and organs, except the digestive organs and urinary bladder, gradually decreased approximately 16 min after the injection. The uptake of radioactivity by the brain is extremely low.

Figure 6 shows the summed PET image (0–90 min) and time–activity curves (TACs) of [^11^C]BPTES. The radioactivity levels in the liver and kidneys peaked at 2.5 min and 1.5 min after injection, and reached 5.9 SUV and 4.2 SUV, respectively. The clearance of radioactivity in the two organs was slow, and the values decreased to 3.2 SUV in the liver and 1.8 SUV in the kidneys at 90 min. After the initial uptake, the radioactivity in the lungs and heart remained at low levels (1.5–2 SUV) until the PET scan. No significant difference was observed in the PET images or TAC values in these tissues after the injection of [^11^C]BPTES without or with unlabeled BPTES (30 mg/kg, intraperitoneal administration) (data not shown). The lack of a decrease in uptake observed could possibly be attributed to the fact that the unlabeled BPTES was injected intraperitoneally one hour prior to the administration of [^11^C]BPTES, which may not have allowed sufficient time for in vivo GLS1 target occupation.

### 2.5. Biodistribution

To further investigate the distribution pattern of [^11^C]BPTES in the whole body, we conducted a biodistribution study using mice. Table 1 shows the radioactive concentrations (% ID/g tissue) of [^11^C]BPTES at five time points in mouse organs and tissues. At 1 min after the injection of [^11^C]BPTES, high uptakes of radioactivity (>5% ID/g) were observed in the blood, heart, lungs, liver, pancreas, small intestine, and kidneys. After the initial uptake, the radioactive uptake showed washout from the blood, heart, lungs, pancreas, and kidneys. The uptake in the liver increased at the initial phase and then decreased from 5 min to 60 min after the injection. The radioactivity level in the small intestine increased with time to reach a maximum value (28.81% ID/g) at 60 min after the injection, suggesting biliary excretion of radioactivity.

### 2.6. Metabolite Analysis

We analyzed the radiolabeled metabolite in mouse plasma and liver to investigate the in vivo stability of [^11^C]BPTES (Table 2). The fraction corresponding to the intact form of [^11^C]BPTES in the plasma decreased to 36% at 15 min, 24% at 30 min, and 11% at 60 min after the injection, while the levels of [^11^C]BPTES in the liver were 73% at 15 min, 62% at 30 min, and 15% at 60 min. The plasma and liver observed more polar radiolabeled metabolites than [^11^C]BPTES. These results suggested that [^11^C]BPTES possessed moderate in vivo metabolic stability. Furthermore, the percentage of the intact form of [^11^C]BPTES was higher in the liver than in the plasma until 30 min after the injection, suggesting that the uptake of radioactivity in the liver may be related to certain in vivo specific bindings of [^11^C]BPTES.

## 3. Materials and Methods

All chemicals and organic solvents were purchased from FUJIFILM WAKO Pure Chemicals (Osaka, Japan), Tokyo Chemical Industry (Tokyo, Japan) or KANTO Chemical (Tokyo, Japan) and used as supplied. ^1^H-NMR and ^13^C-NMR spectra were recorded on a JEOL ECA-500 (500 MHz) spectrometer (JEOL Japan) in the analytical instrumentation center of Chiba University. The chemical shifts were measured as ppm downfield relative to the tetramethyl silane (0 ppm) and DMSO-d_6_ (39.6 ppm) signals, respectively. Signals are quoted as s (singlet); d (doublet); t (triplet); q (quartet); m (multiplet); br (broad). Coupling constants (J values) are given in hertz (Hz). ESI-MS spectra were recorded on a Thermo scientific Q-Exactive Plus spectrometer in the analytical instrumentation center of Chiba University. Melting point (MP) was measured with a Yanaco MP-500V instrument (Kyoto, Japan) and data were uncorrected. Column chromatography was performed using Wako-Gel C-200 (100–200 mesh).

No-carrier-added [^11^C]CO_2_ was produced using a CYPRIS HM-18 cyclotron (Sumitomo Heavy Industry, Tokyo, Japan), through the bombardment of dry N_2_ gas with a beam (15–20 μA) of 18 MeV protons (14.2 MeV on target). High performance liquid chromatography (HPLC) was performed using a JASCO HPLC system (JASCO, Tokyo, Japan). The effluent radioactivity in radio-HPLC was monitored using a NaI (Tl) scintillation detector system. Unless otherwise stated, radioactivity was measured with an IGC-3R Curiemeter (Aloka, Tokyo, Japan).

### 3.1. Chemical Synthesis

#### 3.1.1. 5,5′-(Thiobis(ethane-2,1-diyl)) bis (1,3,4-thiadiazol-2-amine) (**5**)

Phosphorous oxychloride (23 mL, 247 mmol) was added dropwise to a mixture of 3,3′-thiodipropanoic acid (3; 2.67 g, 15 mmol) and thiosemicarbazide (4; 2.73 g, 30 mmol). This reaction mixture was heated at 100 °C for 5 h [19]. After the reaction, the mixture was cooled to room temperature, added dropwise to ice-water, and neutralized to pH > 11 with 5 M NaOH solution. The obtained precipitate was collected by filtration, washed with water and EtOH, and dried in vacuo to give compound 5 (2.32 g, 53.6%) as a white powder. Mp: 221–222 °C. ^1^H-NMR (500 MHz, DMSO-d_6_) 7.05 (s, 4H), 3.07 (4H, t, *J* = 7.2 Hz), 2.85 (4H, *J* = 7.3 Hz). ^13^C-NMR (125.7 MHz, DMSO-d_6_) 169.08, 156.93, 30.78, 30.52. HRMS (ESI) m/z 289.0357 [M + H]^+^ (calcd for C_8_H_13_N_6_S_3_: 289.0358).

#### 3.1.2. N-(5-(2-((2-(5-Amino-1,3,4-thiadiazol-2-yl)ethyl)thio)ethyl)-1,3,4-thiadiazol-2-yl)-2-phenylacetamide (1) and N,N′-((Thiobis(ethane-2,1-diyl))bis(1,3,4-thiadiazole-5,2-diyl))bis(2-phenylacetamide) (BPTES)

Phenylacetylchloride (10 mmol, 1.32 mL) was added dropwise to a mixture of 5 (2.88 g, 10 mmol) and Et_3_N (20 mmol, 2.8 mL) in DMF (250 mL). This reaction mixture was stirred at room temperature for 12 h. After the reaction, the mixture was quenched by water and evaporated in vacuo to remove DMF. The obtained precipitate was washed with water, filtrated, and dried in vacuo. The crude products (a mixture of BPTES and 1) were separated by column chromatography eluted CH_2_Cl_2_/ CH_3_OH (99/1 to 90/10) to yield BPTES and 1 as white powders, respectively.

BPTES: 620 mg (11.8%). Mp: 245–246 °C. ^1^H-NMR (500 MHz, DMSO-d_6_) 7.32–7.35 (8H, m), 7.26–7.28 (2H, m), 3.80 (4H, s), 3.25 (4H, t, *J* = 7.2 Hz), 2.92 (4H, t, *J* = 7.2 Hz). ^13^C-NMR (125.7 MHz, DMSO-d_6_) 169.46, 162.40, 158.65, 134.75, 129.35, 128.54, 126.99, 41.65, 30.19, 29.48. HRMS (ESI) m/z 525.1193 [M + H]^+^ (calcd for C_24_H_25_N_6_O_2_S_3_: 525.1196).

Compound 1: 732 mg (18.0%). Mp: 212–213 °C. ^1^H-NMR (500 MHz, DMSO-d_6_) 7.25–7.36 (5H, m), 7.05 (2H, brs), 3.81 (2H, s), 3.36 (1H, s) 3.25 (2H, t, J = 7.0 Hz), 3.07 (2H, t, *J* = 7.2 Hz), 2.92 (2H, t, *J* = 7.2 Hz), 2.85 (2H, t, *J* = 7.2 Hz). ^13^C-NMR (125.7 MHz, DMSO-d_6_) 169.50, 168.94, 162.44, 158.68, 156.47, 134.76, 129.37, 128.57, 127.02, 41.67, 30.30, 30.22, 30.05, 29.52. HRMS (ESI) m/z 407.0773 [M + H]^+^ (calcd for C_16_H_19_N_6_OS_3_: 407.0777).

### 3.2. Radiosynthesis of [^11^C]BPTES

During the production of [^11^C]CO_2_, benzylmagnesium chloride (BnMgCl, 1 M in THF, 500 μL) was passed through a polyethylene loop (0.75 mm i.d. and 1.6 mm o.d. × 200 mm) cooled between −5 and 0 °C and previously flushed with N_2_ gas [32]. N_2_ gas (50 mL/min) was passed through the loop for 30 s to remove the excess BnMgCl solution and to leave a thin film of BnMgCl on the surface of the loop. After irradiation, [^11^C]CO_2_ was carried from the target with a stream of N_2_ gas and trapped in the stainless-steel coil cooled by liquid N_2_. After trapping of [^11^C]CO_2_, this coil was heated at 50 °C and the released [^11^C]CO_2_ was transferred in an N_2_ stream (3.0 mL/min) into the loop coated with BnMgCl. By passing a solution of ClCOOiBu/iPr_2_NEt/THF (5 μL/5 μL/200 μL) through the loop containing the Grignard’s reaction mixture, the total mixture was transferred into an empty reaction vessel and let stand for 3 min at room temperature to produce the mixed anhydride [11C]2. After the formation of the mixed [^11^C]2, a solution of 1/iPr_2_NEt/NMP (1.5 mg/5 μL/300 μL) was added and heated at 80 °C for 5 min.

After the reaction, the reaction mixture was diluted by the HPLC solvent (1 mL) and applied onto a semipreparative HPLC column. The preparative HPLC conditions were as follows: column, CAPCELL PAK UG80 C_18_ (10 mm i.d. × 250 mm, OSAKA Soda, Osaka, Japan); mobile phase, CH_3_OH/50 mM ammonium acetate / DMSO (70/30/0.1); flow rate, 5.0 mL/min; retention time (t_R_), 7.9 min). The HPLC fraction corresponding to [^11^C]BPTES was collected in a sterile flask containing polysorbate 80 (75 μL), EtOH (0.3 mL), and 25% ascorbic acid (100 μL). All the solvents were removed under reduced pressure. Physiological saline (5 mL) was added in the flask to dissolve the residue. The resulting solution was sterilized using a Millex-GV filter (Millipore) to obtain [^11^C]BPTES as a final product.

HPLC analysis was performed to determine radiochemical purity, identity, and molar activity of [^11^C]BPTES. The analytic conditions were as follows: column, Capcell Pak UG80 C_18_ (4.6 mm i.d. × 250 mm, OSAKA Soda); mobile phase, CH_3_OH/50 mM ammonium acetate/DMSO (70/30/0.1); flow rate: 1.0 mL/min; UV detection, 254 nm; t_R_, 8.3 min. The identity of [^11^C]BPTES was confirmed by co-injection with BPTES. The molar activity was measured and calculated by comparing the assayed radioactivity with the mass measured at UV (254 nm). The logD_7.4_ value of [^11^C]BPTES was measured in an octanol/buffer mixture at room temperature.

### 3.3. Animal Experiments

#### 3.3.1. Animals

Animals were maintained and handled in accordance with the recommendations made by the Committee for the Care and Use of Laboratory Animals at the National Institutes for Quantum Science and Technology (QST). All experiments performed at QST were approved by the Animal Ethics Committee of QST (approval number: 16-1006). C57BL/6 mice were supplied by Japan SLC (Shizuoka, Japan) and housed under a 12/12 h dark/light cycle under optimal conditions.

#### 3.3.2. Small-Animal Pet Imaging

The PET scanning was carried out using a small-animal PET scanner (Siemens Medical Solutions USA, Knoxville, TN, USA). Mice (7–8 weeks old, 30.1 ± 1.4 g, *n* = 3) were anesthetized using 1.5% (*v/v*) isoflurane and treated according to the protocol reported previously [28,29]. Mice were intravenously injected via the tail vein with [^11^C]BPTES (14–16 MBq, 0.3 mL) and a 90 min scan was conducted immediately. The time frame reconstruction was as follows: 1 min × 4 frames, 2 min × 8 frames, and 5 min × 12 frames. The obtained dynamic PET images were reconstructed by filtered-back projection using a Hanning filter, with a Nyquist cutoff of 0.5 cycles per pixel. The time–activity curves (TACs) of [^11^C]BPTES were acquired using PMOD software (version 3.4, PMOD Technology, Zurich, Switzerland) from the volumes of interest, which were manually mapped onto the liver, kidney, lung, brain and heart. The radioactivity was decay-corrected to the injection time and presented as a standardized uptake value (SUV), which was normalized for injected radioactivity and body weight. The SUV was calculated as follows: SUV = (radioactivity per milliliter tissue/injected radioactivity) × body weight (g). All PET quantitative data are expressed as the mean ± standard deviation (SD).

#### 3.3.3. Biodistribution Study

Mice (8 weeks old, 31.9 ± 0.7 g) were intravenously injected via the tail vein with [^11^C]BPTES (2.7 MBq, 0.1 mL) and were sacrificed by cervical dislocation at 1, 5, 15, 30, and 60 min (*n* = 3 at each time point) after the injection. The blood samples (heart contents) were collected, and the whole brain, heart, thymus, lung, liver, pancreas, spleen, kidneys, adrenal gland, stomach (including contents), small intestine (including contents), large intestines (including contents), muscle, bone, testis, and bladder were quickly harvested and weighed. The radioactivity level in each tissue was measured using the automatic gamma counter (2480 Wizard^2^, PerkinElmer, MA, USA) and was expressed as percentage of the injected dose per gram of wet tissue (% ID/g). Decay correction was accounted for during the radioactivity measurements.

#### 3.3.4. Metabolite Analysis

Mice (8 weeks old, 32.0 ± 0.3 g) were intravenously injected via the tail vein with [^11^C]BPTES (37 MBq, 0.3–0.5 mL) and were then sacrificed by cervical dislocation at 15, 30, and 60 min (*n* = 3 for each time point) after the injection. Blood and liver samples were obtained quickly and treated according to the procedure reported previously [37]. An aliquot of the supernatant (0.2–1.0 mL) obtained from the plasma or liver homogenate was injected into the HPLC system with a radioactivity detector, and analyzed using a Capcell Pak C_18_ column (4.6 mm i.d. × 250 mm) with a mobile phase of CH_3_OH/50 mM ammonium acetate/DMSO (70/30/0.1) at a flow rate of 1.0 mL/min. The percentage ratio of [^11^C]BPTES (t_R_ = 8.3 min) to total radioactivity (corrected for decay) on the HPLC chromatogram was calculated as % = (peak area for [^11^C]BPTES/total peak area) × 100.

## 4. Conclusions

In the present study, [^11^C]BPTES was successfully synthesized for the first time through the reaction of [^11^C-carbonyl]acylation of amine precursor 1 with [^11^C]2, which was prepared by the Grignard reaction of benzyl magnesium chloride with [^11^C]CO_2_, followed by treatment with isobutyl chloroformate in situ. Moreover, a PET study with [^11^C]BPTES showed high radioactivity uptake in the liver, kidney, and small intestine. The distribution pattern of [^11^C]BPTES throughout the entire body was further confirmed through ex vivo biodistribution analysis. Metabolite analysis revealed moderate metabolism of [^11^C]BPTES in vivo. To the best of our knowledge, this report represents the first investigation into the pharmacokinetics of BPTES in living animals. Recent studies have reported that the antitumor and senolytic effects of BPTES and its analogs and NPs are rapidly increasing [14,21,22,23,24,25]; however, their in vivo mechanisms are not yet fully understood, partly due to the poor bioavailability of BPTES. PET with [^11^C]BPTES could serve as a tool to better understand the tissue distribution, kinetics of uptake and clearance, and differential effects related to treatment, such as single-dose versus pre-loading. This understanding will improve our knowledge of the kinetics of BPTES for the further development of the glutaminase inhibitor, BPTES analogs, and BPTES-NPs.

## Figures and Tables

**Figure 1 pharmaceuticals-16-00963-f001:**
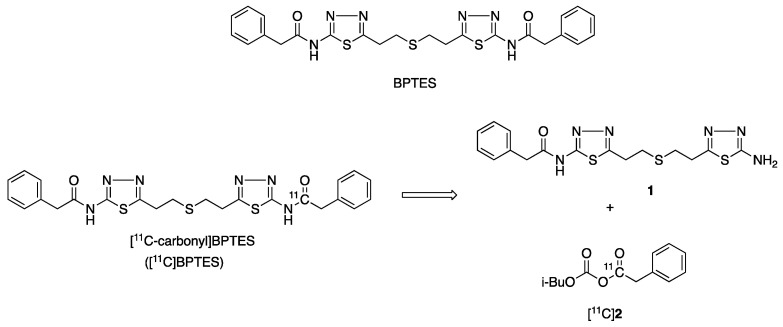
Chemical structures of BPTES and [^11^C-carbonyl]BPTES and retrosynthesis of [^11^C]BPTES.

**Figure 2 pharmaceuticals-16-00963-f002:**
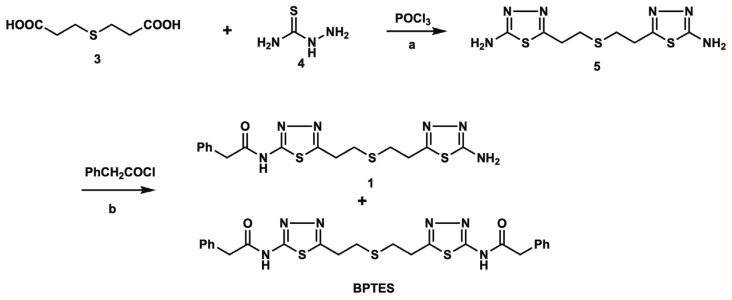
Chemical synthesis of the labeling precursor **1** and BPTES: a. 100 °C, 5 h, 53.6%; b. Et_3_N, room temperature, 12 h, 18.0% for **1**, 11.8% for BPTES.

**Figure 3 pharmaceuticals-16-00963-f003:**
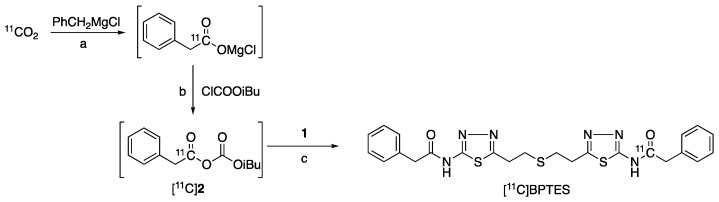
Radiosynthesis of [^11^C]BPTES: a. 1 min, −5–0 °C; b. iPr_2_N-Et, NMP, room temperature, 1 min; c. NMP, 80 °C, 5 min.

**Figure 4 pharmaceuticals-16-00963-f004:**
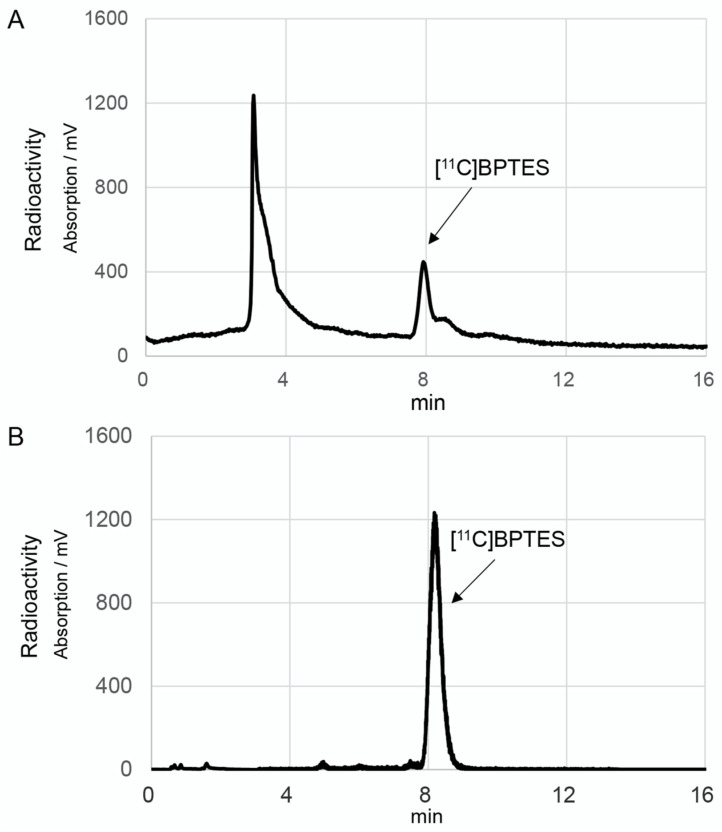
HPLC separation (**A**) and analytical (**B**) charts of [^11^C]BPTES. The HPLC separation conditions were as follows: CAPCELL PAK UG80 C_18_ column (10 mm i.d. × 250 mm length, Osaka Soda, Osaka, Japan), CH_3_OH/50 mM Ammonium acetate/DMSO (70/30/0.1), 5.0 mL/min. The HPLC analytic conditions were as follows: CAPCELL PAK UG80 C_18_ column (4.6 mm i.d. × 250 mm length, OSAKA Soda), CH_3_OH/50 mM ammonium acetate/DMSO (70/30/0.1), 1.0 mL/min.

**Figure 5 pharmaceuticals-16-00963-f005:**
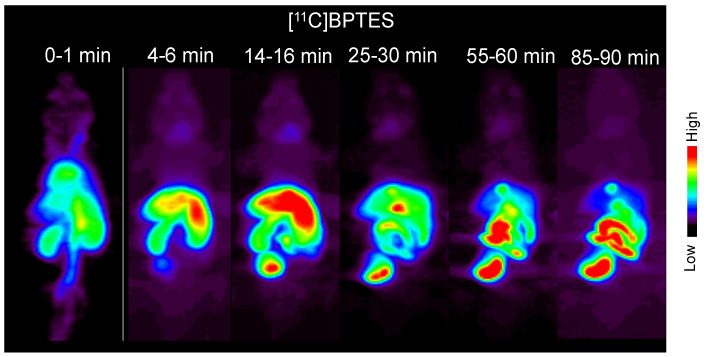
Typical static PET images of whole body of a mouse at different time points after the injection of [^11^C]BPTES. PET scans were performed on three C57BL/6 mice.

**Figure 6 pharmaceuticals-16-00963-f006:**
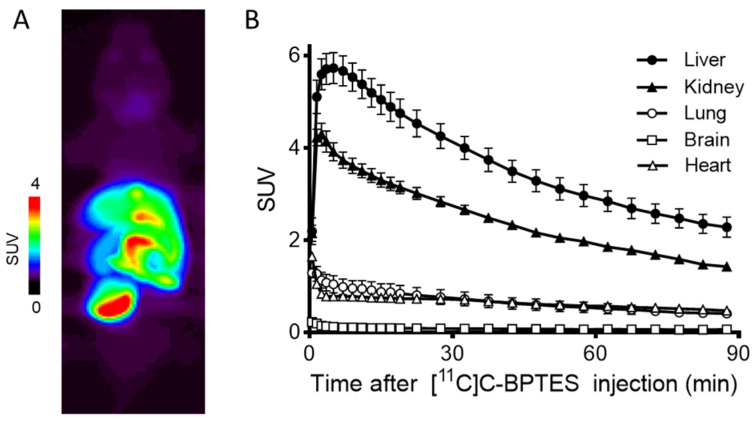
Representative PET image (**A**) and time–activity curves (TACs) (**B**) of [^11^C]BPTES in C57BL/6 mice (*n* = 3). Radioactivity was detected and expressed as SUV. The SUV value (mean ± SD, *n* = 3) was calculated according to the following formula: measured activity concentration (Bq/mL) × body weight (g)/injected activity (Bq).

**Table 1 pharmaceuticals-16-00963-t001:** Biodistribution of [^11^C]BPTES in C57BL/6 mice.

Organ	1 min	5 min	15 min	30 min	60 min
Blood	6.12 ± 0.12	1.61 ± 0.08	1.4 ± 0.17	1.06 ± 0.2	0.47 ± 0.03
Heart	5.11 ± 0.16	3.56 ± 0.5	2.86 ± 0.2	2.07 ± 0.05	1.69 ± 0.16
Thymus	1.86 ± 0.03	0.88 ± 0.05	0.92 ± 0.1	0.69 ± 0.15	0.44 ± 0.12
Lung	18.77 ± 3.95	6.04 ± 0.2	2.83 ± 0.2	1.56 ± 0.06	1.23 ± 0.07
Liver	27.85 ± 4.3	34.74 ± 1.35	31.46 ± 0.86	20.96 ± 1.9	12.96 ± 1.55
Pancreas	5.86 ± 0.6	5.34 ± 0.22	4.84 ± 0.22	3.66 ± 0.09	2.66 ± 0.31
Spleen	4.05 ± 0.71	3.03 ± 0.64	1.38 ± 0.07	0.86 ± 0.08	0.69 ± 0.13
Kidneys	40.73 ± 3.06	33.19 ± 2.59	19.84 ± 1.04	12.22 ± 0.22	7.09 ± 0.57
A. gland	4.54 ± 0.92	3.05 ± 0.59	1.53 ± 0.1	1.74 ± 0.23	1.04 ± 0.19
Stomach	2.9 ± 0.37	2.5 ± 0.49	2.68 ± 0.21	6.92 ± 1.89	10.4 ± 4.22
S. intestine	10.59 ± 1.79	17.02 ± 3.25	28.16 ± 6.72	24.9 ± 5.26	28.91 ± 10.12
L. intestine	2.53 ± 0.79	2.19 ± 0.42	1.71 ± 0.27	2.4 ± 0.23	7.29 ± 1.81
Muscle	0.78 ± 0.03	0.89 ± 0.09	0.93 ± 0.07	0.71 ± 0.06	0.62 ± 0.08
Bone	1.56 ± 0.19	1.32 ± 0.23	0.85 ± 0.06	0.57 ± 0.13	0.35 ± 0.09
Testis	0.33 ± 0.11	0.35 ± 0.01	0.5 ± 0.16	0.41 ± 0.1	0.09 ± 0.01
Bladder	3.94 ± 2.74	5.2 ± 1.09	8.95 ± 1.71	5.56 ± 0.86	12.97 ± 10.89
Brain	0.68 ± 0.05	0.36 ± 0.02	0.19 ± 0.01	0.14 ± 0.01	0.1 ± 0.01

Data are %ID/g tissue (mean ± SEM; *n* = 3). A. gland: Adrenal gland; S. intestine: small intestine; L. intestine: Large intestine.

**Table 2 pharmaceuticals-16-00963-t002:** Percentages of intact form of [^11^C]BPTES in the plasma and liver of C57BL/6 mice.

Time after Injection	Plasma	Liver
15 min	36.11 ± 1.68	73.13 ± 6.28
30 min	23.71 ± 2.55	62.33 ± 4.68
60 min	11.14 ± 3.00	14.50 ± 7.85

Data are expressed as % of tailored BPTES entity (mean ± SEM; *n* = 3).

## Data Availability

Data is contained in the article.

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
