# Peer review of "The Glutaminase-1 Inhibitor [11C-carbony]BPTES: Synthesis and Positron Emission Tomography Study in Mice"

_pharmaceuticals, 2023, doi:10.3390/ph16070963_

Round 1
Reviewer 1 Report
The study describes the radiosynthesis of [11C]BPTES and its in vivo biodistribution using PET, for the first time. As a proof-of-concept, the study provides valuable insights into senolytic effects of BPTES and the new BPTES analogs.
The study is well written, the methods are suffienntly clear descrived and the results are correctly presented.
I recommend to be published in present form.
Author Response
Response: We sincerely appreciate Reviewer 1 for summarizing our work and for investing your time and effort in reviewing our manuscript. We are also pleased that the reviewer 1 recommends our manuscript for publication in Pharmaceuticals.

Reviewer 2 Report
The authors report on the first radiolabeling of BPTES, as selective inhibitor of glutaminase-1, with C-11 to assess its pharmakokinetic behavior through PET studies. The work is comprehensive and well done. The molecule shows relatively high liver uptake, which is not surprising given the hydrophobicity of the lead structure. Here is my first question: I miss the determination of a logD7.4 value. I recommend the authors to measure the logD7.4 value for [11C]C-BPTES, as this is fundamental in medicinal chemistry and would provide an explanation for the in vivo behavior.
My second question is on the blocking experiments in vivo. The authors state that no significant difference in PET images or TAC values were observed, if unlabeled BPTES was used. In theory that means that the compound does not bind selectively to any target. The authors should add a brief comment or explanation for why no blocking effect was observed in the "results and discussion" section.
A third point is the nomenclature of radiolabeled compounds. I strongly recommend to follow the guidelines of the international radiopharmaceutical community reported in H. H. Coenen et al., Nuclear Medicine and Biology 2017, Vol. 55 Pages v-xi. It has to be written [11C]C-BPTES.
A minor language mistake in lines 131 and 132. Please correct the sentence to: ...,mice were subjected to PET scans. Remove what is written afterwards.
English language is overall fine. Some minor editing of typos is suggested as mentioned above in the comments.
Author Response
We sincerely appreciate reviewer 2 for summarizing our work and providing thoughtful and constructive comments on our manuscript. We have carefully considered all of your suggestions and have made the necessary revisions to the manuscript accordingly. We hope that you will find it suitable for the final acceptance and publication in Pharmaceuticals after revision. Below is our point-by-point response to your comments.
